# Sicilian Populations of *Capparis spinosa* L. and *Capparis orientalis* Duhamel as Source of the Bioactive Flavonol Quercetin

**DOI:** 10.3390/plants12010197

**Published:** 2023-01-03

**Authors:** Francesco Sgadari, Antonietta Cerulli, Rosario Schicchi, Natale Badalamenti, Maurizio Bruno, Sonia Piacente

**Affiliations:** 1Department of Agricultural, Food and Forest Sciences (SAAF), Università degli Studi di Palermo, Viale delle Scienze, ed. 4, 90128 Palermo, Italy; 2Department of Pharmacy, University of Salerno, 84084 Fisciano, Italy; 3Department of Biological, Chemical and Pharmaceutical Sciences and Technologies (STEBICEF), Università degli Studi di Palermo, Viale delle Scienze, ed. 17, 90128 Palermo, Italy; 4Centro Interdipartimentale di Ricerca “Riutilizzo bio-Based Degli Scarti da Matrici Agroalimentari” (RIVIVE), Università di Palermo, 90128 Palermo, Italy

**Keywords:** *Capparis spinosa*, *Capparis orientalis*, Capparaceae, quercetin, LC-ESI/QTrap/MS/MS, pedoclimatic features

## Abstract

The genus *Capparis* is a taxon of difficult delimitation that has several species and ecotypes due to its wide heterogeneity, its extreme phenotypic diversity, and the presence of intermediate forms linked to hybridization phenomena. The Sicilian territory hosts numerous wild and cultivated populations of two spp. *Capparis spinosa* L. and *Capparis orientalis* Duhamel, which are ecologically and morphologically distinct. The caper has considerable interest and economic value for its medicinal properties, culinary uses, and cultivation characteristics. It is one of the foods with the highest quercetin content. Quercetin is a flavonol with antioxidant, anti-inflammatory, and immunostimulant properties. Recently, patents and clinical studies have highlighted the inhibitory effect of this compound against several SARS-CoV-2 enzymes (MPro, PLPro, and RdRp). Therefore, the aim of this study was to quantify the amount of quercetin in *C. spinosa* and *C. orientalis* by LC-ESI/QTrap/MS/MS and to correlate it with the pedoclimatic features. The results obtained showed that quercetin is more abundant in *C. orientalis* than in *C. spinosa*. The highest values of quercetin were recorded in *C. orientalis* flowers, leaves, and flower buttons of volcanic islands with southwest and east warm exposures. In conclusion, the data acquired can provide a good basis for further scientific investigations to support the identification of possible ecotypes as a source of quercetin for food or pharmaceutical purposes.

## 1. Introduction

*Capparis* is a nano-phanerophyte genus with a Eurasian-subtropical chorotype, belonging to the family Capparaceae [1]. The genus *Capparis* L. has about 250–300 species, arboreal, shrubby, lianose, and herbaceous [2], widespread across tropical and subtropical regions of the world (World Flora Online) and cultivated in North Africa, southern Europe, central and southern Asia, and Australia. *Capparis* is a heliophilous and xerophilous genus, with a highly developed root system, and with a high root/stem ratio [3,4], which allows it both to reduce soil erosion and slow down the desertification process [5], as well as high water-use efficiency, so irrigation requirements are very limited; indeed, irrigations are only carried out in the first year of planting [6]. These characteristics make this crop suitable for maintaining the sustainability of agroecosystems threatened by global warming [4].

The taxonomic classification of the genus *Capparis* has undergone several revisions over the years, considering the extreme phenotypic diversity and the presence of intermediate forms related to hybridization phenomena [7]. In particular, taking into consideration the individuals present in the Sicilian territory (Italy), there are only two species, *Capparis spinosa* L. and *Capparis orientalis* Duhamel (World Flora Online), which are distinct from an ecological, morphological, and genetic point of view, both known as caper.

The various parts of the caper plants are rich mainly in phenolic acids, flavonoids, alkaloids, phytosterols, sugars, vitamins, and organic acids [8,9,10]; among these, the main flavonoids are quercetin, kaempferol, and their derivatives. Quercetin is regarded as one of the most powerful antioxidants among polyphenols [11,12], but it is known also for its anti-inflammatory, antihistamine, anticarcinogenic, antiviral, antioxidant, and psychostimulant properties [13,14,15,16,17,18]. Moreover, the importance of quercetin has recently increased, demonstrating that it can act as a specific inhibitor for the virus responsible for COVID-19 as it has a destabilizing effect on 3CLpro, one of the key proteins for viral replication [19]. Precisely, in consideration of the biological activity exerted by quercetin and the renewed importance attributed to this flavonol in the prevention and treatment of SARS-CoV-2 [19,20,21], the objective of this research is to define the amount of this flavonol in the epigean parts of the two species of *Capparis* occurring as Sicilian and *circum*Sicilian populations, by liquid chromatography coupled to tandem mass spectrometry with the electrospray (ESI) source and the hybrid triple quadrupole-linear ion trap mass analyzer (LC-ESI/QTrap/MS/MS), working in multiple reaction monitoring (MRM) mode. The work, developed in several phases, was aimed at obtaining data on the growth areas and on the distribution of the two species of *Capparis* present in Sicily, considering the possible chemo-taxonomic differences, and correlating the amount of quercetin in the epigean parts (flowers, leaves, branches, flower buttons, and fruits) to the pedoclimatic features at which the different populations collected are subjected.

## 2. Results

With the aim to quantify quercetin in the epigean parts of *C. spinosa* and *C. orientalis*, 22 stations falling within both Sicily and the *circum*sicilian islands were selected, taking into account the different stationary characteristics, as reported in Table 1 and Table 2 for *C. spinosa* and *C. orientalis*, respectively.

To define the amount of quercetin in the selected species, LC-ESI/QTrap/MS/MS analysis, using the MRM mode, has been carried out. MRM is a selective and sensitive spectrometric technique in which a specific transition from a precursor ion to a product ion is monitored [22,23]. Quercetin displayed a pseudomolecular ion [M-H]^−^ at *m/z* 301, which was characterized by a diagnostic fragmentation at *m/z* 151, which was derived by the Retro-Diels–Alder reaction (RDA) [24]; consequently, this key transition was chosen for MRM analysis. Based on this transition, the amount of quercetin (µg/g dry plant) in extracts obtained from *C. spinosa* and *C. orientalis* was determined (Figure 1, Table 3 and Table 4).

The results of the chemical analyses performed on the twenty-two populations showed that there are significant quantitative differences between the two species, among the different parts of the plant, and among the different sampling stations. In fact, as can be deduced from Table 3, *C. spinosa* populations contain a lower average amount of quercetin in all parts of the plant than *C. orientalis* (Table 4). The lowest amount of quercetin was found in branches and fruits of either species, with average values of 5.5 and 7.7 (μg of quercetin per g of dry plant), while the highest values were recorded in flowers, leaves, and flower buttons, with average values of 114.4, 94.0, and 90.7 μg/g, respectively, for both species. Regarding the sampling stations, the highest value was found in the flower buttons of sample **11**, amounting to 908.0 μg/g. This value differs significantly both from those recorded at the other stations, including also the two samples of the same island (**12** and **13**), and from those of the various parts of the plant. A high quantity of quercetin was also found in the leaves of the different sampling stations, and in particular, sample **14** recorded the highest amount of 787.0 μg/g, followed by sample **11,** with a value of 264.0 μg/g. Moreover, it was observed that plants vegetating in hill stations (**1**, **3**, **4**, **5**, and **7**) had quercetin values ranging from 19.0 to 128.0 μg/g in flowers, from 7.0 to 135.0 μg/g in leaves, and from 3.0 to 55.0 μg/g in flower buttons. Samples on coastal stations (**2**, **6**, **8**, **9**, and **10**), instead, had quercetin quantities ranging from 14.0 to 333.0 μg/g in flowers, from 17.0 to 144.0 μg/g in leaves, and from 12.0 to 37.0 μg/g in the flower buttons. Comparing plants present in the smaller islands (**11**, **12**, **13**, **14**, **15**, **16**, **17**, **18**, **19**, **20**, **21**, and **22**), quercetin values from 7.0 to 393.0 μg/g were detected in flowers, especially in stations with northern (**13, 20**) and western (**22**) exposures, from 4.0 to 908.0 μg/g in flower buttons, and from 5.0 to 787.0 μg/g in leaves, especially in populations with southwest (**11**, **14**) and northern (**19**, **20**) exposures.

In leaves of island populations, the highest concentration was found in calcareous soils with an amount of 787.0 μg/g (**14**), followed by the value of sample **11** (264.0 μg/g) on volcanic soil. In flowers, instead, the amount of quercetin was higher in volcanic soils (**13**, **22**) than in calcareous soil (**14**, **17**, and **18**).

## 3. Discussion

This work investigated the biodiversity of the genus *Capparis* in Sicily to determine the quercetin amount present in the various epigean parts of the plants. In the Sicilian territory, there are only two species that are ecologically, morphologically, and genetically distinct. *C. spinosa* is widespread, mainly in xerophilous areas on regosols and lithosols and on other Miocene substrates of sedimentary origin such as marls and clays [25], while *C. orientalis* is widespread in rocky environments, cliffs, and crags near the coast; on Mesozoic and Oligocene compact limestones; on marls, on chalks, and on substrates of volcanic origin. The study was conducted on 22 stations within both Sicily and the *circum*sicilian islands. For this purpose, sampling was carried out under different growth conditions and at various phenological stages, acquiring information on the taxa collected, their distribution, and their possible cultivation techniques.

The chemical analyses carried out showed that quercetin is mainly present in flowers, leaves, and flower buttons, although there were discrepancies in the results of the analyses at the different sampling stations, which could be due to the impossibility of collecting the plant materials on the same day and, consequently, standardizing the climatic conditions before collection. These results agreed with those present in the work of Moghaddasian et al. [26] carried out in Iran, although they differed in the absolute values of quercetin present in individual epigean parts. The compounds present in the caper plants are strongly influenced by the geographical and environmental conditions, harvest period, storage methods, genotype, and method of extraction and processing [27]. These factors can have significant effects on the determination of the final antioxidant profile and the medicinal properties of the species. 

The differences highlighted in flower buttons collected in Pantelleria, especially between **11** (908.0 μg/g) and **12** (114.0 μg/g), could be due both to the fact that **11** corresponds to spontaneous individuals of *C. orientalis*, while **12** is a caper grove submitted during each growing season to different fertilization and insecticide treatments, and to the fact that the two varieties grow at a different altitude. On the other hand, the differences found between **11** (908.0 μg/g) and **13** (180.0 μg/g) can be due to both different exposure and, in the case of sample **13**, to a greater distance from the sea. In addition, it is important to point out that the data obtained does not take into account the difference in caliber existing between the flower buttons harvested. This parameter, as pointed out in the work of Giuffrida et al. [28], influenced the average amount of flavonoids whose synthesis is greater in the smaller and younger flower buttons since these compounds exert a protective and defensive action for the plant in the growing parts. 

The results highlighted that the concentration of quercetin present in flowers, leaves, and flower buttons also seems to be correlated with the stress factors to which the plants are subjected: exposure, altitude, distance from the sea, and soil characteristics. Therefore, it is possible to affirm that, in flowers, the higher amount of quercetin is related to geographical sites with higher humidity, whereas, generally, in leaves and flower buttons it is related to drier and more arid sites. Furthermore, the quantity of quercetin is correlated to the type of soil on which plants grow; in fact, in flower buttons, a higher quantity of quercetin was found in volcanic soils, particularly in the stations on the islands located in the southwest of Sicily (**11**, **12**, **13**, **15**, and **16**), with the highest values found in the flower buttons of samples **11** (908.0 μg/g) and **15** (245.0 μg/g), subjected to southwest and east warm exposures. 

## 4. Materials and Methods

### 4.1. Plant Materials 

*C. spinosa* and *C. orientalis* populations were sampled from a wide area of the Sicilian territory, between May and September 2021. Plant material was obtained from 22 different populations (5 for *C. spinosa* and 17 for *C. orientalis*), choosing an average of over 30 specimens representative of the stand based on morphological characteristics of individual plants. The following parts were collected from each individual plant: flowers, leaves, branches, flower buttons, and fruits. Sampling sites include the provinces of Trapani and Palermo through the “Gessoso-Solfifera” series to the Agrigento area, the Ibleo plateau, and the province of Messina; including the Pelagie Islands, Pantelleria, Favignana, Marettimo, Ustica, and Salina islands (Figure 2). 

Contextually, qualitative characteristics concerning leaf morphotype and stipules were analyzed, which allowed, according to the classification elaborated by Gristina et al. [29], for the determination of the sampled species.

The collected samples include natural and cultivated populations present on different substrates: volcanic, calcareous, chalky, and clayey. The sampling location, site description and substrate type, primary (cliffs, clays) and secondary habitat (semi-rural environment, dry stone walls, old roadside walls, and railroad ballast), exposure and altitude above sea level, and geographic (GPS) coordinates and slope were recorded during the sampling phases (Table 1 and Table 2). The different samples, identified by Prof. Rosario Schicchi, have been stored in the University of Palermo Herbarium (Voucher No. 110657-110678).

### 4.2. Extraction of Plant Material

For each population, flowers, leaves, branches, flower buttons, and fruits, weighed in the field using a technical scale (~100 g), once frozen (−20 °C), were subjected to a freeze-drying process (ScanVac CoolSafe 15L Freeze Dryers, LABOGENE, Bjarkesvej 5, DK-3450 Allerød, Denmark) until they reached a constant weight. Subsequently, the different lyophilized parts were extracted following the procedure reported by Moghaddasian et al. (2012) [26]. Therefore, 500 mg of plant material was taken from each sampled part, ground, and then subjected to an extraction process for 24 h at room temperature with a methanol-acetic acid-water solution at a ratio of 100:2:100. After 24 h, the solution was filtered, reduced in volume, lyophilized again until a constant weight was reached, and subsequently subjected to LC-ESI/QTrap/MS/MS analysis.

### 4.3. LC-ESI/QTrap/MS/MS Analysis

Quantitative analysis was performed on an LC-ESI/QTrap/MS/MS system, operating in MRM mode, on a C18 reversed-phase (RP) column (50 mm × 2.1 mm; Luna Omega C18 1.6 µm; Phenomenex, Aschaffenburg, Germany) kept at 30 °C, using water as phase A and acetonitrile as phase B, both with 0.1% of formic acid, at a flow rate of 0.3 mL/min. The autosampler was set to inject 3 μL of each sample (1.0 mg/mL). Each extract was analyzed in negative ion mode, and a linear gradient was used; in particular, starting from 5% B and increasing to 15% B in 1.0 min, successively to 35% B in 3.9 min, from 35% B to 95% in 2 min, returning to 5% B in 1 min, and finally held at 5% B for 5 min. Linearity was evaluated by correlation values of calibration curves. The limit of quantification (LOQ; equivalent to sensitivity) was evaluated by injecting a series of increasingly diluted standard solutions until the signal-to-noise ratio was reduced to 10. The limit of detection (LOD) was estimated by injecting a series of increasingly diluted standard solutions until the signal-to-noise ratio was reduced to 3 [30]. LOD was 0.0003 ng/mL, and LOQ was 0.001 ng/mL.

The stock solution (1 mg/mL) of quercetin was used as an external standard (ES). This solution was diluted with methanol to obtain nine solutions of different ES concentrations (0.001, 0.05, 0.1, 0.25, 0.5, 1.0, 2.5, 5.0, and 7.0 µg/mL). Moreover, the following parameters were settled for quercetin: a declustering potential (DP) of −60 eV, a focusing potential (FP) of −354 eV, an entrance potential (EP) of −5 eV, a collision energy (CE) of 30%, and a collision cell exit potential (CXP) of −30 eV. In this way, a calibration curve, analyzed by linear regression (y = 0.00372x − 0.0987, R2 = 0.995), by Analyst 1.6.2 software provided by the manufacturer (AB Sciex), was obtained for quercetin. An appropriate amount of internal standard (IS; resveratrol) was added to yield a final concentration of 1.0 ng/µL. For calibration curves, 3 μL of each standard solution at each concentration level in triplicate was used. The ratio of the peak area of the ES to those of the IS was calculated and plotted against the corresponding concentrations of the standard compounds using weighted linear regression to generate standard curves.

## 5. Conclusions

In this work, two different species, belonging to the genus *Capparis*, were analyzed: *C. spinosa* and *C. orientalis*. In particular, the amount of quercetin in the various epigean parts of the plant (flowers, leaves, branches, flower buttons, and fruit) was investigated by LC-ESI/QTrap/MS/MS analysis. The amount of this secondary metabolite was also correlated with the pedoclimatic characteristics of the twenty-two sampling sites examined in Sicily and in the *circum*sicilian islands. The data showed that *C. orientalis* contained significantly more quercetin than *C. spinosa*, a species not present on the islands around Sicily. Relative to *C. orientalis*, quercetin was more abundant (245 to 908 μg/g) in flower buttons of volcanic islands with southwest and east warm exposures. On the other hand, in leaves of these populations, the highest quantity was found in calcareous soils (787.0 μg/g) (**14**), followed by the value of sample **11** (264.0 μg/g) on volcanic soil. In flowers, instead, the amount of quercetin was higher in both volcanic soils (**13**, **22**) than in calcareous soil (**14**, **17**, and **18**).

In conclusion, the data acquired can provide a good basis for further scientific investigations to support the identification of possible ecotypes as a source of quercetin for food or pharmaceutical purposes. This could be a major boost for the socio-economic development of rural areas in arid or semi-arid regions of Mediterranean environments since the caper is a species of remarkable interest.

## Figures and Tables

**Figure 1 plants-12-00197-f001:**
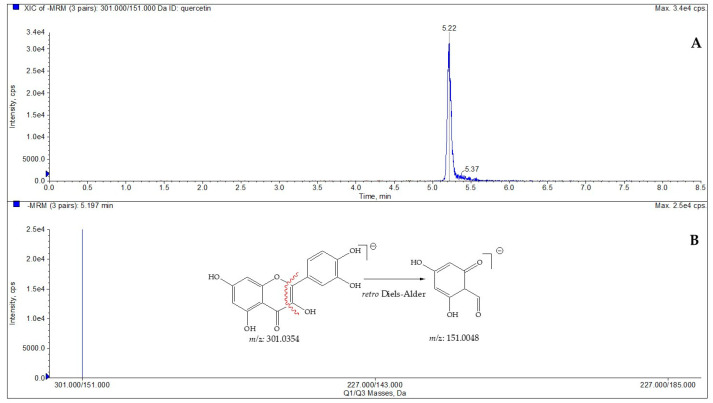
LC-ESI/QTrap/MS/MS spectrum of quercetin. The representative extracted ion chromatogram (XIC) of multiple-reaction monitoring (MRM) chromatogram of quercetin (**A**) and MRM MS spectrum of quercetin with the diagnostic fragmentation at *m/z* 151.0048 (**B**).

**Figure 2 plants-12-00197-f002:**
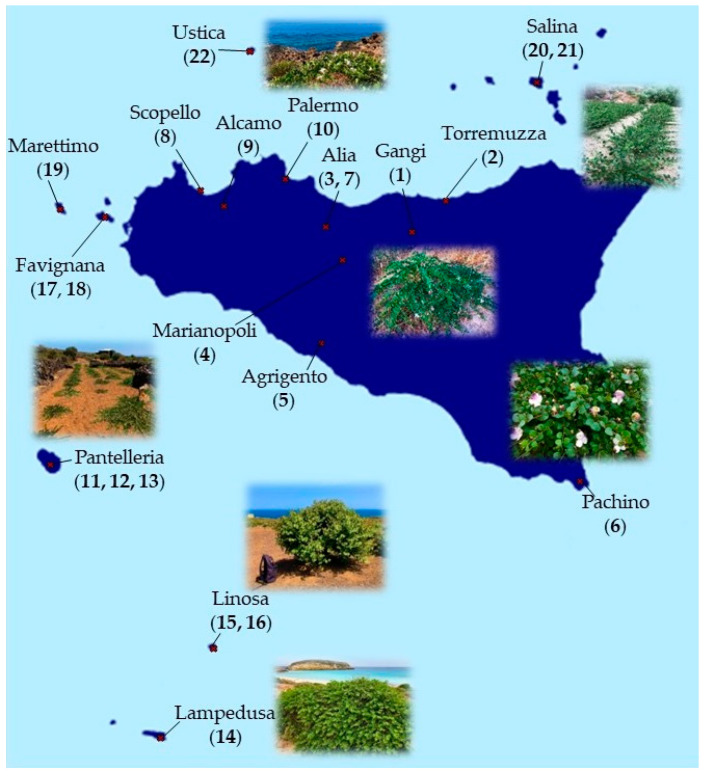
Geographical distribution of the different populations of *Capparis* collected.

**Table 1 plants-12-00197-t001:** *C. spinosa* sampling site description.

Sample	Station	Sampling Site Description
**1**	Casalgiordano, Gangi, Palermo	Edges of the SP14 Gangi-Casalgiordano road on regosols, southern exposure, 590 m a.s.l., slope 20/40° (37°43′35″ N 14°11′24″ E)
**2**	Ex oleificio Giannì, Torremuzza, Messina	Coastal Villa Margi-Torremuzza on flat land, southwest exposure, 10 m a.s.l. (38°0′40″ N 14°19′26″ E)
**3**	Contrada Chianchitelle, Alia, Palermo	Roadside near the Luigi Sturzo Technical Institute, southwest exposure, 630 m a.s.l., on flat terrain (37°46′5″ N 13°43′45″ E)
**4**	Train station, Marianopoli, Caltanissetta	Massive railway SS121, northeast exposure, 310 m a.s.l., 20° slope (37°37′18″ N 13°53′56″ E)
**5**	Valle dei Templi, Agrigento, Caltanissetta	Margin cultivated fields on limestone tuff rocks, southwest exposure, 90 m a.s.l., 90° slope (37°17′45″ N 13°35′15″ E)

**Table 2 plants-12-00197-t002:** *C. orientalis* sampling site description.

Sample	Station	Sampling Site Description
**6**	Spiaggia Morghella, Pachino, Syracuse	Edges of SP84 road on limestone tuff rocks, southern exposure, 20 m a.s.l., slope 10–20° (36°42′16″ N 15°7′16″ E)
**7**	Alia, Palermo	Old road retaining walls, southern exposure, 650 m a.s.l., slope 30–90° (37°46′41″ N 13°42′50″ E)
**8**	Riserva dello Zingaro, Scopello, Castellamare del Golfo, Trapani	Trail margin on lithosols near the nature museum, east exposure, 30 m a.s.l., 10/20° slope (38°5′11″ N 12°48′25″ E)
**9**	Contrada Fico, Alcamo, Trapani	Rocky tuff limestone wall on SS113, northwest exposure, 170 m a.s.l., 90° slope (37°59′29″ N 13°0′18″ E)
**10**	Botanical Garden, Palermo	Giardino dei Semplici, northeast exposure, 20 m a.s.l., on level ground (38°6′44″ N 13°22′20″ E)
**11**	Contrada Scauri, Pantelleria, Trapani	Roadside on volcanic rocks, southwest exposure, 70 m a.s.l., 20° slope (36°45′2″ N 11°58′55″ E)
**12**	Contrada Nicà, Pantelleria, Trapani	Cappereto azienda la Nicchia, west exposure, 170 m a.s.l., on level ground (36°45′22″ N 11°59′6″ E)
**13**	Lago di Venere, Pantelleria, Trapani	Road edges on volcanic rocks, northern exposure, 50 m a.s.l., slope 20/40° (36°49′10″ N 11°59′18″)
**14**	Spiaggia dei conigli, Lampedusa, Agrigento	Trail margin on calcarenites, southwest exposure, 20 m a.s.l., slope 0/20° (35°30′50″N 12°33′26″E)
**15**	Faraglioni, Linosa, Agrigento	Trail margin on volcanic rocks, east exposure, 10 m a.s.l., slope 0/30° (35°51′51″ N 12°52′52″ E)
**16**	Cala Mannarazza, Linosa, Agrigento	Road edges on volcanic rocks, northern exposure, 50 m a.s.l., slope 10/30° (35°52′26″ N 12°52′35″ E)
**17**	Cala Azzurra, Favignana, Trapani	Calcarenitic rocky ridge, southeast exposure, 10 m a.s.l., slope 20° (37°54′34″ N 12°21′41″ E)
**18**	Castello di Santa Caterina, Favignana, Trapani	Roadside margin on calcarenitic rocks, east exposure, 100 m a.s.l., slope 10° (37°55′39″ N 12°19′3″ E)
**19**	Spiaggia De Rotolo, Marettimo, Trapani	Roadside margin on lithosols with rock outcrops, northeast exposure, 10 m a.s.l., slope 50/80° (37°57′42″ N 12°4′38″ E)
**20**	Frazione Malfa, Salina, Messina	Cappereto with northern exposure, 60 m a.s.l., slope 10–20° (38°34′42″ N 14°50′14″ E)
**21**	Frazione Pollara, Salina, Messina	Roadside on volcanic rocks, southwest exposure, 60 m a.s.l., 90° slope (38°34′53″ N 14°48′26″ E)
**22**	Caletta Acquario, Ustica, Palermo	Margin of the road on volcanic rocks, west exposure, 10 m a.s.l., slope 0/20° (38°42′10″ N 13°9′26″ E)

**Table 3 plants-12-00197-t003:** Quercetin content (μg) in the epigean parts of *C. spinosa*.

*Capparis spinosa*					
Sample	μg Quercetin/g Dry Plant				
	Flowers	Leaves	Branches	Flower Buttons	Fruits
**1**	44.0	135.0	13.0	55.0	13.0
**2**	14.0	44.0	9.0	37.0	18.0
**3**	19.0	14.0	2.0	3.0	3.0
**4**	128.0	7.0	4.0	18.0	2.0
**5**	22.0	7.0	1.0	7.0	2.0

**Table 4 plants-12-00197-t004:** Quercetin content (μg) in the epigean parts of *C. orientalis*.

*Capparis orientalis*					
Sample	μg Quercetin/g Dry Plant				
	Flowers	Leaves	Branches	Flower Buttons	Fruits
**6**	35.0	19.0	9.0	25.0	22.0
**7**	121.0	21.0	14.0	29.0	15.0
**8**	281.0	144.0	2.0	12.0	9.0
**9**	333.0	21.0	6.0	12.0	3.0
**10**	274.0	17.0	1.0	15.0	7.0
**11**	7.0	264.0	15.0	908.0	4.0
**12**	9.0	5.0	6.0	114.0	2.0
**13**	277.0	93.0	2.0	180.0	3.0
**14**	33.0	787.0	1.0	178.0	3.0
**15**	66.0	5.0	3.0	245.0	4.0
**16**	4.0	6.0	1.0	35.0	3.0
**17**	159.0	5.0	3.0	39.0	5.0
**18**	72.0	38.0	10.0	24.0	7.0
**19**	10.0	185.0	3.0	4.0	7.0
**20**	169.0	230.0	6.0	14.0	5.0
**21**	46.0	11.0	3.0	16.0	14.0
**22**	393.0	11.0	8.0	25.0	20.0

## Data Availability

Not applicable.

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
