# Peer review of "Sicilian Populations of Capparis spinosa L. and Capparis orientalis Duhamel as Source of the Bioactive Flavonol Quercetin"

_plants, 2023, doi:10.3390/plants12010197_

Round 1

Reviewer 1 Report

kindly provide the n LC-ESI/QTrap/MS/MS  spectra of quercitin

2. kindly draw the structure of quercitin

Reviewer 2 Report

The general theme of the article „Sicilian populations of Capparis spinosa L. and Capparis orientalis Duhamel as source of the bioactive flavonol quercetin“ is of interest. The manuscript is clearly laid out. All key elements, Abstract, Introduction, Results, Discussion, and Material and Methods, are present. The title clearly describes the article. The manuscript is well written.

 I have noted the following minor errors that require the attention of the authors:

Line 78: correct „respectivel“ to respectively and add full stop

Line 81: please explain what the sign * means

In Table 1 and 2 please explain the meaning of the abbreviations given in round brackets next to the name of the station, such as (PA), (ME), (CA) and so on.

According to the Instructions for Authors of the journal: „Acronyms/Abbreviations/Initialisms should be defined the first time they appear in each of three sections: the abstract; the main text; the first figure or table. When defined for the first time, the acronym/abbreviation/initialism should be added in parentheses after the written-out form.“

Line 141: The reference „La Bella et al. 2021“ is not cited in the text by numbers placed in square brackets as it should be. That reference, entitled „Four-Year Study on the Bio-Agronomic Response of Biotypes of  Capparis spinosa L. on the Island of Linosa (Italy)“,  should be included to the reference list.

Lines 144 – 148: The sentence is too long and not intelligible enough. It should be emphasized that the results discussed refer to flower buttons.

Lines 242 – 244: The sentence is a bit vague. I suggest deleting the word „still“. Also, the last sentence in the Abstract section is good, and can also be used in conclusion too.

Maybe as follows:

„In conclusion, it is believed that the data obtained provide a good basis for further scientific investigations to identify ecotypes that can be introduced into cultivation and serve as a source of quercetin for food or pharmaceutical purposes.“

In summary, I believe that the submitted manuscript is suitable for publication in Plants.
